# Orthohantaviruses, Emerging Zoonotic Pathogens

**DOI:** 10.3390/pathogens9090775

**Published:** 2020-09-22

**Authors:** Emmanuel Kabwe, Yuriy Davidyuk, Anton Shamsutdinov, Ekaterina Garanina, Ekaterina Martynova, Kristina Kitaeva, Moffat Malisheni, Guzel Isaeva, Tatiana Savitskaya, Richard A. Urbanowicz, Sergey Morzunov, Cyprian Katongo, Albert Rizvanov, Svetlana Khaiboullina

**Affiliations:** 1Institute of Fundamental Medicine and Biology, Kazan Federal University, 420008 Kazan, Russia; emmanuelkabwe@ymail.com (E.K.); davi.djuk@mail.ru (Y.D.); shamsutdinov2006@yandex.com (A.S.); kathryn.cherenkova@gmail.com (E.G.); ignietferro.venivedivici@gmail.com (E.M.); olleth@mail.ru (K.K.); rizvanov@gmail.com (A.R.); 2Kazan Research Institute of Epidemiology and Microbiology, 420012 Kazan, Russia; guisaeva@rambler.ru (G.I.); tatasav777@mail.ru (T.S.); 3Ministry of Health, Zambia, Lusaka 10101, Zambia; malishenitasheni@gmail.com; 4Wolfson Centre for Global Virus Infections, University of Nottingham, Nottingham NG7 2UH, UK; richard.urbanowicz@nottingham.ac.uk; 5School of Life Sciences, University of Nottingham, Nottingham NG7 2UH, UK; 6Department of Pathology, School of Medicine, University of Nevada, Reno, NV 89557, USA; 7Department of Biological Sciences, University of Zambia, Lusaka 10101, Zambia; katongo1960@yahoo.co.uk; 8Department of Microbiology and Immunology, University of Nevada, Reno, NV 89557, USA; sv.khaiboullina@gmail.com

**Keywords:** hemorrhagic fever with renal syndrome, hantavirus pulmonary syndrome, Puumala orthohantavirus, nephropathia epidemica, reassortment, recombination, emerging, zoonosis, HFRS and HPS

## Abstract

Orthohantaviruses give rise to the emerging infections such as of hemorrhagic fever with renal syndrome (HFRS) and hantavirus pulmonary syndrome (HPS) in Eurasia and the Americas, respectively. In this review we will provide a comprehensive analysis of orthohantaviruses distribution and circulation in Eurasia and address the genetic diversity and evolution of Puumala orthohantavirus (PUUV), which causes HFRS in this region. Current data indicate that the geographical location and migration of the natural hosts can lead to the orthohantaviruses genetic diversity as the rodents adapt to the new environmental conditions. The data shows that a high level of diversity characterizes the genome of orthohantaviruses, and the PUUV genome is the most divergent. The reasons for the high genome diversity are mainly caused by point mutations and reassortment, which occur in the genome segments. However, it still remains unclear whether this diversity is linked to the disease’s severity. We anticipate that the information provided in this review will be useful for optimizing and developing preventive strategies of HFRS, an emerging zoonosis with potentially very high mortality rates.

## 1. Introduction

Hemorrhagic fever with renal syndrome (HFRS) and hantavirus pulmonary syndrome (HPS) are zoonotic infections commonly diagnosed in Eurasia and the Americas, respectively. HFRS is caused by several orthohantaviruses including Hantaan (HTNV), Seoul (SEOV), Puumala (PUUV), and Dobrava-Belgrade (DOBV), while infection with Sin Nombre (SNV) and Andes (ANDV) orthohantaviruses are linked to HPS [1,2]. It is believed that orthohantaviruses are transmitted by inhaling aerosolized virus, direct contact, or via rodent bite [3]. Person-to-person transmission has only been demonstrated for ANDV [4]. The mortality rate of HPS and HFRS varies substantially, reaching 60% and 12%, respectively [5]. A mild form of HFRS, also known as nephropathia epidemica (NE), is endemic in Europe and has the lowest mortality rate of 0.4% [1,2]. Of all European countries, the highest prevalence of HFRS is documented in Russia, where more than 117 000 cases have been recorded since the year 2000, resulting in 516 fatalities. These cases also include 2800 children under the age of 14 [6].

Several orthohantaviruses have been shown to circulate in Eurasia [7,8]. However, the most common ones associated with disease in humans include HTNV, SEOV, PUUV, and DOBV. HTNV and SEOV mainly circulate in Asia, with many cases documented in China and South Korea, where the primary hosts are *Apodemus agrarius* and *Rattus norvegicus*, respectively. SEOV has also been identified in rats globally [9]. DOBV has been identified in *Apodemus flavicollis, A. agrarius*, and *A. ponticus*, and mainly circulates in the Balkans, Russia, and Denmark [9,10]. In Europe, DOBV is the most pathogenic to human with many lineages identified with various degree of virulence [11,12].

The most commonly identified orthohantavirus causing nephropathia epidemica (NE), a mild form of HFRS, in European countries is PUUV [13]. PUUV is usually detected in bank voles, whose habitat extends from Spain to Siberia and from the Balkans to Northern Scandinavia [13,14]. The forest and forest-steppe zones that cover vast territories in Europe support the bank vole population by providing food and shelter [15]. The bank vole migrations contribute to the spread of PUUV as natural hosts entering new areas. When bank vole rodent population peaks, PUUV reinfection and infection of bank voles could occasionally happen, adding to virus genetic diversity [16]. Moreover, contact between different infected groups of rodents leads to the exchange of PUUV strains through re- and co-infection [17].

PUUV strains can be modified as the result of mutations and exchange of the genome segments when viruses infect new hosts [17]. Mutations can generate many distinct PUUV genome variants and lead to the formation of new strains. Currently, eight PUUV genetic lineages are identified in the bank voles: Central European (CE), Alpe-Adrian (ALAD), Danish (DAN), South Scandinavian (S-SCAN), North Scandinavian (N-SCAN), Finnish (FIN), Russian (RUS) and the newly identified Latvian (LAT) [18,19,20,21,22,23] (Table 1). It was demonstrated that the nucleotide genetic diversity between PUUV lineages for the S (small) segment is 15–19% [24,25], while the differences in nucleotide sequences between PUUV strains within the same lineage could range from 0 to 9% [20,26].

The PUUV genetic diversity results from both the accumulation of the point mutations and rearrangement within the viral genome, such as recombination and reassortment [23,30]. While the molecular evolution of orthohantaviruses can lead to the emergence of strains that are more adapted to the rodent host, it is believed that the strain’s genome variations could lead to the substantial changes in the virus pathogenicity in humans [30]. Moreover, mutations in the PUUV genome could complicate the identification of the PUUV strain involved in the outbreak [31,32] as diagnosis of PUUV infections in humans includes real-time RT-PCR based methods [31,33]. Multiple commercial kits were compared by Reynes et al., who identified several mismatching primers and probes to the genome of PUUV strains circulating in France [32], which could have serious implications in detecting (and missing) possible infections. Therefore, careful studies of PUUV genome variations have fundamental and practical importance for epidemiological and medical control. Understanding the processes leading to genome diversity will help with the prediction of PUUV variants and assessing their emerging infection potentials to cause severe disease and outbreaks.

Genetic recombination and reassortment most likely take place in what is known as “contact zones” between adjacent rodent populations [34] that are infected with different strains. This is common in bank vole’ communities as co-circulation of two PUUV strains within a limited area have been reported in Finland [34], Sweden [29], Latvia [18] and Russia [35]. However, the analysis of the PUUV diversity in European regions and others still remains limited. To address this gap of knowledge, we aimed to provide a comprehensive review of the orthohantaviruses distribution and circulation in Eurasia. Moreover, we will address genetic diversity and the evolution of PUUV causing HFRS in this region, which is an important emerging infection.

## 2. Genome Structure of Orthohantaviruses

According to the recent classification by the International Committee on Taxonomy of Viruses 2019, orthohantaviruses belong to the genus Orthohantavirus, and family *Hantaviridae* (https://talk.ictvonline.org/taxonomy/). Orthohantaviruses are enveloped spherical viruses, 80–160 nm in diameter [36,37]. The lipid-containing shell has spikes formed by viral glycoproteins (Gn and Gc) used for attachment to the target cell [37,38]. Orthohantavirus’ genome is single-stranded, negative-sense RNA, containing three segments: S (small), M (medium) and L (large) which code for the nucleocapsid (N) protein, envelope glycoproteins (Gn and Gc), and the RNA-dependent RNA polymerase (RdRp), respectively [37,39] (Figure 1). Additionally, the S segment of some (PUUV and TULV) orthohantaviruses consists of an overlapping open reading frame, which codes for a non-structural protein (NSs) [40].

N protein is the most abundant viral protein synthesized early after infection [41]. This protein plays a role in intracellular transportation and assembly of mature virions [3,42]. In addition, N protein facilitates the attachment of the virus to the host cell proteins promoting replication [3]. Moreover, the NSs protein regulates the interferon response in infected cells by inhibiting IFN-ß promoter [43]. Gn and Gc are directly involved in binding to the target cell receptors and entry [3,44]. Moreover, glycoproteins can modulate the immune response [38]. RdRp mediate replication and transcription of the orthohantavirus genome [3,45]. To do this, through unknown mechanisms, they first synthesize cRNA, which is used to produce the viral S, M and L mRNAs [3,45].

Orthohantavirus S, M, and L segments are approximately 1828, 3650, and 6550 nucleotides (nt) long, respectively [39,46]. PUUV genome includes the coding sequences as well as non-coding regions (NCR) located at the end of 5′ and 3′ (5′ NCR and 3′ NCR, respectively) (Figure 2). The length of the coding sequences is approximately 1299, 3444, and 6468 nt for S, M, and L segments, respectively. For all orthohantavirus genomes, the size of the NCR at the 5′ end is similar for each segment, 42 nt for the S, 40 nt for M, and 36 nt for the L [46]. Due to the high number of nucleotide insertions and deletions, the length of NCR at the 3′ end differs even for the most closely related viral strains [47]. In addition, the overlapping open reading frame in the S segment genome of PUUV and TULV orthohantaviruses is 270 nt long [43,48].

The length of the 3′ NCR is approximate ~320 nt, ~280 nt, and ~120 nt for the S, M, and L segments, respectively [21,46]. The NCR short sequence 3′-AUCAUCAUCUG, which is complementary to the 5′ ends, is located at the end of each segment of all orthohantaviruses.

## 3. Distribution of Orthohantaviruses and the Diseases They Cause

Orthohantaviruses are classified according to the geographic distribution of their hosts and the diseases they cause. Currently, more than 30 genetically distinct orthohantaviruses have been described in small animals and birds worldwide [21], which are divided into two groups: Old and New World orthohantaviruses. It is proposed that orthohantaviruses are spread horizontally between rodent and insectivore populations, with zoonotic spillover into humans occurring when they come into contact with aerosols from contaminated animal droppings (Figure 3). The main factors influencing the dynamics and distribution of the animal host population includes climate conditions such as humidity, precipitation, and temperature [30,49]. Junyu et al. found positive correlation between the humidity and precipitation with high incidence of HFRS in Changsha city, China [49]. It appears that the favorable climate led to a high production of crop supporting reproduction of rodents due to the abundance of food [49,50]. Other factors impacting rodent population includes the land covers such as shrubs, grassland, or chaparral with many green plants providing food and shelter for the rodents [49]. Knowledge of the climate and environmental characteristics of the region is critical to develop appropriate health strategies for HFRS control and prevention.

The Old and New World orthohantaviruses cause HFRS and HPS, respectively [5,38]. However, it is important to note that HFRS virus-specific antibodies have also been detected in residents in Argentina, Brazil, Canada, Colombia, and the USA, countries where HPS is endemic [51]. Moreover, this study revealed the anti-HFRS antibodies in Alaska and Hawaii Islands population [51]. The world distribution of pathogenic orthohantaviruses is summarized in Figure 4. These results could indicate the high cross reactivity between different hantaviruses [51].

HFRS is endemic in Asia, mostly in China, North Korea, and South Korea, with 90% of all cases reported in China [49,52,53]. The most common causative agents of HFRS in Asia are HTNV and SEOV. Although significant intervention measures involving rodent control and infection preventions were implemented, HFRS remains a serious public health problem in China [52].

Moreover, the disease has been diagnosed in Europe (Sweden, Norway, Finland, Bulgaria, Slovenia, Czech Republic, Belgium, France, and UK) [54,55,56]. Additionally, the serological studies revealed the presence of orthohantavirus specific antibodies in humans in other European countries [57]. In Slovenia, Poland, and the Czech Republic, DOBV is the orthohantavirus most commonly associated with HFRS diagnosis [7], while PUUV is mainly detected in the former Yugoslavia, Sweden, Finland, Germany, Belgium, France, and Poland [7,58]. Tula (TULV) orthohantavirus was first isolated from common European voles (*Microtus arvalis* and *M. rossiaemeridionalis*) captured in the central part of Russia in 1987 [21]. Later, TULV was also isolated from the vole samples collected in the central European countries [59]. A novel vole-associated hantavirus related to TULV and PUUV, Tatenale virus (TATV), has been identified in field voles (*Microtus agrestis*) captured in different parts of England [60,61]. This virus is believed to be non-pathogenic in humans and rodents [62].

The highest incidence and prevalence of orthohantavirus infection is documented in Russia, Finland, and Sweden [63]. In Russia, many cases of HFRS are registered in the Volga region (Bashkiria, Udmurtia, and Tatarstan), the southern part of Siberia (Omsk, Tyumen, and Novosibirsk), and the Far East region (Amursky, Primorsky, and Khabarovsky regions) [63]. PUUV is the most commonly identified orthohantavirus causing HFRS in these areas, while only sporadic infections exist for DOBV [14,25,64]. Usually, each orthohantavirus is associated with specific natural host rodent and their geographical location [65,66,67] (Table 2, also see Figure 4).

In addition to PUUV, DOBV, and TULV, several other orthohantaviruses were isolated in Russia and linked to HFRS: HTNV, SEOV, Amur, Artybash, and Altay [94]. Moreover, a new orthohantavirus identified in the Siberian lemming (*Lemmus sibiricus*) population in the Taimyr Peninsula, Russia, was named Topografov virus [47]. Additionally, Khabarovsk orthohantavirus was isolated in the Khabarovsk and Primorsky Krai, Far-East, Russia [27]. SEOV, first discovered in the Republic of Korea, has now spread throughout the Eurasian countries and detected in rodents captured in the Far East of Russia [70] (Figure 4).

Orthohantaviruses are spread around the world, where their distribution strongly correlates with the host areas providing the potential of several emerging zoonoses.

## 4. Gene Exchange between Orthohantaviruses

New strains of segmented viruses, like orthohantaviruses, are often the result of the genetic reassortment and recombination [89]. This occurs when one cell is infected with two different strains of the same virus or two different species of orthohantaviruses; the progeny could have a mixed genome from both parental strains (Figure 5).

### 4.1. Recombination

Recombination is intra-molecular switch of templates in RNA viruses or non-homologous or homologous in both RNA and DNA viruses [95]. Virus genome recombination depends on the frequency of the genome segments exchange between two different parent strains and the rate of co-infection of the same host cell [96]. The frequency of recombination has been estimated in vitro. Based on Froissart et al. [95], on average, more than 50% of the in vitro produced viral genomes of cauliflower mosaic virus were recombinants. Analysis of the recombination rate showed that this process affected all regions of the genome. The authors stated that ten cycles of viral replication are required to produce the recombinant progeny. The recombination frequency per base and per replication cycle was 2 × 10^−5^ to 4 × 10^−5^ [95].

In the *Hantaviridae*, genome recombination has been reported in vitro and in vivo [97]. In contrast to other viruses, the frequency of genome recombination in the *hantaviridae* family members is not fully understood [59]. Natural recombination between genetically closely related strains was shown to occur relatively often, primarily if these strains circulate in the same rodent population [28]. For example, a recombination mechanism was suggested for the generation of some PUUV strains belonging to N-SCA, S-SCA, and DAN lineages [28]. In HTNV, SEOV, ANDV, TULV, and PUUV, recombination was demonstrated in the S segment, while M segment recombination was described only in HTNV in China [98].

Due to the limited understanding of recombination in nature, the association between recombination and orthohantavirus pathogenesis is still under investigation. Moreover, the effect of recombinant variants of the virus on the immune response remains mostly unknown.

### 4.2. Reassortment

Reassortment is the exchange of the genome segments between viruses that co-infect the same cell. This event requires a high degree of compatibility at the level of RNA and proteins [11]. It is believed that reassortment played a role in the emergence of the segmented viruses through the complementation process [11]. This exchange of the genome is one of the factors contributing to the molecular evolution of orthohantaviruses and to the currently recognized orthohantavirus diversity [11,99].

Natural reassortment was demonstrated between SNV strains in 1994 [100] as well as La Crosse and Tahyna viruses in 1991 [101]. Since then, many cases of natural reassortment between segmented viruses of the same and different lineages were reported [34]. Razzauti et al. were the first to describe natural PUUV genome segments’ reassortment isolated from the bank voles of the parental strains, although it still remains to be determined if they can successfully compete with the parental strains in vivo [23]. In another study, 19.1% of PUUV genomes obtained from rodents in 2005–2009 were identified as reassortants [18]. Of these reassortants, the majority had S or M segments exchanged, while the L segment always remained linked with the parental S or M segments [18].

For the replication process of segments from two parental genomes in one cell, no differences in replication are observed when infected with one strain but the packaging process that generates virus progenies with some genes from both parents appears to be not accidental. In a number of studies, the packaging of S and L segments of the same parental strain together with the M segment of the other strain was demonstrated in PUUV [102,103]. This could be due to the critical role that the M segment plays in infection of the cells [104]. In other orthohantavirus such as ANDV and SNV reassortment was experimentally confirmed in several in vitro studies [103,105]. The reassortment between ANDV and SNV, closely related viruses, was 8.9%, where S and L segments were from SNV, while the M segment was from ANDV [105]. The analysis revealed that the reassortants have the replication efficacy similar to ANDV, which was the most effective in in vitro propagation [105]. This data suggests that the new reassortant could retain the highest efficacy of replication, characteristic to one of the parental strains.

Thus, reassortment is one of the mechanisms leading to orthohantaviruses diversity. Identification of the ancestral strains for the natural reassortants could be difficult, especially where one or both of the parental strains are also reassortants. Since the rearrangements of the genetic material of orthohantaviruses can potentially lead to changes in their antigenic and biologic properties, understanding of recombination and reassortment mechanisms is one of the most critical tasks in the studying of HFRS pathogenesis.

## 5. Orthohantavirus Evolution

Originally, orthohantaviruses were isolated from small rodents belonging to the three main subfamilies: *Murinae, Arvicolinae*, and *Sigmodontinae* [106]. However, a study by Bennett et al. has shown that orthohantaviruses can switch from one rodent species to a non-rodent host [66]. It is therefore suggested that as well as the small mammal host, other animals such as, shrews, moles, and bats, occupying the same ecological niche could also carry the viruses [80]. For example, Seewis (SWSV) and Thottapalayam orthohantaviruses were detected in different shrew species [89]. Moreover, several shrew species were identified as natural hosts for orthohantaviruses, including *Sorex araneus*, *S. daphaenodon, S. undress,* and *Neomys anomalus* [87,107]. Several species within the orthohantavirus genus, which are associated with different rodent subfamilies, are represented on the phylogenetic tree (see Figure 6).

Several hypotheses on the origin and evolution of orthohantaviruses were developed in the last decade. According to the co-evolution hypothesis, orthohantaviruses infected their natural host, adapted, and evolved together [35,47]. This hypothesis was supported by the work of Ramsden et al. and Sironen et al. who demonstrated that the S and M genome segments evolved slowly with the evolution rate between 0.7 × 10^−7^ to 2.2 × 10^−6^ and 3.7 × 10^−7^ to 8.7 × 10^−7^ subs/site/year, respectively [20]. This is significantly lower when compared to other RNA viruses, where the evolution rate of 10^−2^ to 10^−4^ nt subs/site/year was shown [108]. These data support the results of the phylogenetic analysis based on a full-length viral genome and rodent mitochondrial DNA (mtDNA) sequences [109]. The phylogenetic tree analysis revealed the parallel evolution of orthohantaviruses and their hosts of *Murinae, Arvicolinae, Neotominae,* and *Sigmodontinae* families [24,109].

Moreover, in the last twenty years, orthohantaviruses have been isolated from insectivores, chiropterans, and other groups of animals and insects. Therefore, the commonly accepted orthohantavirus reservoirs may not be the exclusive natural hosts. In the evolutionary hypothesis proposed by Yanagihara et al. [106], it was suggested that the original natural host of orthohantaviruses could have been an insect that switched host and adapted to a new animal species. Supporting this hypothesis are the data on phylogenetic analysis obtained from investigations of the partial genome S segment nucleotide sequences of Cricetidae species and their association with orthohantaviruses [109]. The results revealed differences in phylogenetic topologies, which can be explained by orthohantaviruses evolving when switching from the natural host, which differs from the co-evolution hypothesis [109]. Moreover, the topological differences in the phylogenetic tree are impacted by natural selection, which includes negative and positive selection forces. Negative selection forces curb the spreading of less fit strains, where the positive selection results in the release of more fit strains, making them an important mechanism of orthohantavirus divergence [110].

There is also an alternative hypothesis suggested explaining orthohantavirus evolution. Ramsden et al., indicated that orthohantaviruses diverged from the single rodent-associated viruses millions of years ago and switched host and adapted to the new environment. This hypothesis is based on the fact that the evolution rate for PUUV, DOBV, and TULV is 2.10 × 10^−2^ to 2.66 × 10^−4^ subs/site/year, which is typical for the evolution rate of mutations in RNA viruses [111]. In contrast, Souza et al. suggest that orthohantaviruses have a recent origin [112]. Based on the calculation of the mutation rate and phylogenetic tree analysis, authors indicated that the common ancestor of all orthohantaviruses could be located approximately 2000 years ago in eastern Asia (presumably in southeast China) [112]. It was recently, 500–700 years ago, when viruses switched from *Murinae* insectivorous species to *Arvicorinae* [112]. This led to the adaptation to the new reservoir, which resulted in the emergence of the new strains of the virus. Subsequently, new strains of orthohantaviruses spread to the Old World and North America, together with their natural hosts [112]. In the New World, orthohantaviruses associated with *Neotominae* rodent species, around 500–600 years ago, reaching North and South America. In Brazil, about 400 years ago, the most recent forms of orthohantaviruses emerged, by adapting to the *Sigmodontinae* rodent species, which spread throughout South America [112]. This model contradicts the long process of co-evolution of orthohantaviruses and their natural hosts, suggested by Dekonenko et al. [35].

The study of orthohantaviruses phylogeny and evolution reveals a large number of orthohantavirus species, which are adapted to a wide range of host animals. The geographical distribution and migration of the rodent hosts could geographically isolate the individual strains of viruses supporting their independent evolution. The multiple different hypotheses on the evolution of orthohantaviruses indicate how difficult it is to study these pathogens, and their asymptomatic activity on the survival of the host may profoundly influence their evolution.

## 6. Genetic Diversity of Orthohantaviruses

The Old World orthohantaviruses such as HNTV, SEOV, DOBV, PUUV, and TULV are all genetically distinct from each other [47]. PUUV is one of the most studied orthohantaviruses of all. Phylogenetically, HNTV forms nine lineages based on the partial M segment sequences, while SEOV forms six lineages and PUUV eight [113]. Analysis of the partial M segment sequences showed that the genetic variability among HTNV strains is higher than that of SEOV strains [114]. In the M segment, the genetic diversity of the HTNV strains identified within a relatively small area displays high nucleotides similarity of about 88% among lineages while when the virus is isolated in different countries, the sequence similarities drop to 85% as it was shown in the case of Chinese and Korean HTNVs [114]. In contrast, the amino acid (aa) sequences of the Chinese and Korean HTNV isolates are conserved with aa similarities of 97%. Another orthohantavirus, DOBV, is represented by two distinct lineages (DOBV-Af and DOBV-Aa) found in different rodent reservoirs [115]. The genetic distances calculated for the strains within one family show higher similarities; however, the genetic diversity between two DOBV lineages reaches up to 14%. Moreover, the diversity increases with increasing geographical separation [115].

### Genetic Diversity of Puumala Orthohantaviruses

Phylogenetically, PUUV is closely related to non-pathogenic viruses TULV and Prospect Hill orthohantavirus (PHV) circulating in Europe and the USA, respectively [47]. Analysis of PUUV sequences revealed a high level of diversity, which form eight genetic lineages [18,24] (Table 1). It was suggested that such diversity of PUUV could be the result of the isolation of the bank vole population in the glacial refuge during the last glacial period (28–23 thousand years ago) [116]. It is believed that these viruses recolonized Eurasia at the end of the ice age [116]. This assumption is supported by orthohantavirus phylogenetic analysis, demonstrating early separation of PUUV genetic lineage ancestors, after which they evolved independently [117].

Of all PUUV genome segments, the S segment is the most studied [20,28]. The large part of 3′NCR of the S segment is characterized by high diversity while the last 100 nt at the 3’end of 3′NCR remains similar in different PUUV strains [20]. The variations in the coding region are up to 19% between lineages, and it is even higher (up to 27%) in the complete S segment [20]. The N and NSs proteins coded by the S segment are 433 aa and 90 aa long [48]. The identity of N protein is 94–100% and for the NSs goes up to 70% between PUUV lineages [48]. The N protein contains the major antigenic site located between 7 aa and 94 aa and a hyper variable region between 232 aa and 275 aa [24]. In some viruses, the nucleotide sequence identity of the S segment shows high diversity above the identity limit accepted for PUUV lineages [24,65,118]. This could be explained by the accumulation of the silent point mutations in the PUUV genome. Moreover, it appears that the missense mutations are under intense selection pressure and represent relatively small changes in N protein aa sequences [20]. Additionally, recombination could lead to PUUV strains diversity. For example, PUUV recombination was demonstrated in the S segment [20] and recombination has been shown in other orthohantaviruses [119]. Although homologous recombination is lower in the PUUV microevolution, it was shown that recombination of the S segments is one of the mechanisms generating virus genetic diversity [23].

It has been proposed that the nucleotide sequence diversity rate is linked to the geographic location of the bank vole population [38,120], where the genetic diversity of PUUV strains found in one territory can significantly differ due to the co-circulation of two lineages within a limited area [18]. Within the bank vole population, the S segment nucleotide diversity of PUUV strains is between 0% and 5%; however, it could reach up to 9.7%, as it was shown for Konnevesi strains [23,47]. When the virus circulates in multiple populations of small rodents, like viruses of RUS and S-SCA lineages, the nucleotide sequence diversity between strains could be up to 15% [23]. For the NSs-ORF nucleotide sequence in the S segment, they are highly conserved with little diversity among the local PUUV strains, like strains of the CE lineage [48]. The NSs-ORF nucleotide sequence diversity among the PUUV strains of different genetic lineages showed high divergence at 35% [48]. The migration of virus-carrying small rodents could contribute to the higher assortment of PUUV strains [20].

PUUV M segment nucleotide identity between different genetic lineages ranges from 80% to 93% [118], while the character within the same family is higher, reaching 90–100% [121]. In some viruses, the M segment sequence has high diversity that exceeds the limits for the inter-lineage definition [118]. For example, the nucleotide sequences dissimilarity between RUS lineage strains CG1820, Kazan, and Samara is 15–16% [21]. This range of variety is characteristic of PUUV sequences from different families [47]. The glycoprotein coded by the M segment has a high aa sequence identity (89–100% and 99–100%) between inter-lineage and intra-lineage sequences, respectively [47,121]. Interestingly, aa sequence differences in Gn and Gc proteins for some PUUV strains reach 7%, which is higher than suggested by the international taxonomy between different orthohantaviruses [122]. This could indicate the circulation of two distinct groups of viruses at the sample location, and the individual voles could be found with a complex mixture of closely related variants of viral strains. Moreover, a 7% difference in aa sequences of Gn and Gc is used in the classification of orthohantavirus species [47,122].

The full coding nucleotide sequences of the L segment from different PUUV lineages have 81–87% identity [118]. Nucleotide sequences of PUUV strains within the same genetic lineage have an even higher identity for the L segment, reaching 91–100% [47,118]. The RdRp, coded by the L segment, is 2156 amino acids long and has 93.4% to 96.1% identity among PUUV strains of the same lineage [47,118]. RdRp aa sequences similarities between PUUV lineages are up to 94% [47,118]. This high similarity in RdRp sequence could indicate the important role of this protein in virus replication, so the nucleotide sequence fidelity is essential to preserve the aa sequence accuracy.

Collectively, the genome (S, M, and L segment) nucleotide sequence diversity of PUUV between different genetic lineages is in the range of 11–19% [22]. This diversity suggests that PUUV could be the most variable virus within the orthohantavirus genus [123], although this could be resultant of sampling bias. It appears that the M segment is the most variable, while the L segment is less variable as compared to the S segment of the PUUV genome. Although significant achievements have been made in identifying the PUUV genetic diversity in Eurasia, more investigations are required to map the circulation of PUUV genotypes in Eurasian countries completely.

## 7. Conclusions

Orthohantaviruses are characterized by a high level of genetic diversity. Accumulation of point mutations and reassortment, which happen in the genome RNA segments, are the main mechanisms leading to new genome variants. Currently, there is no consensus regarding the evolution rate, the origin of orthohantaviruses, and the co-evolution with their rodent/insectivore hosts. Of all orthohantaviruses, high genome diversity is characteristic for PUUV: The identity difference between nucleotide sequences in some parts of the genome, even within the same genetic lineage, can reach 15%, which exceeds the accepted diversity level for the genomic difference. However, it remains unclear whether this diversity is linked to disease severity. Continued research into this virus genus is of great importance as its potential as an emerging infection has been demonstrated throughout the world.

## Figures and Tables

**Figure 1 pathogens-09-00775-f001:**
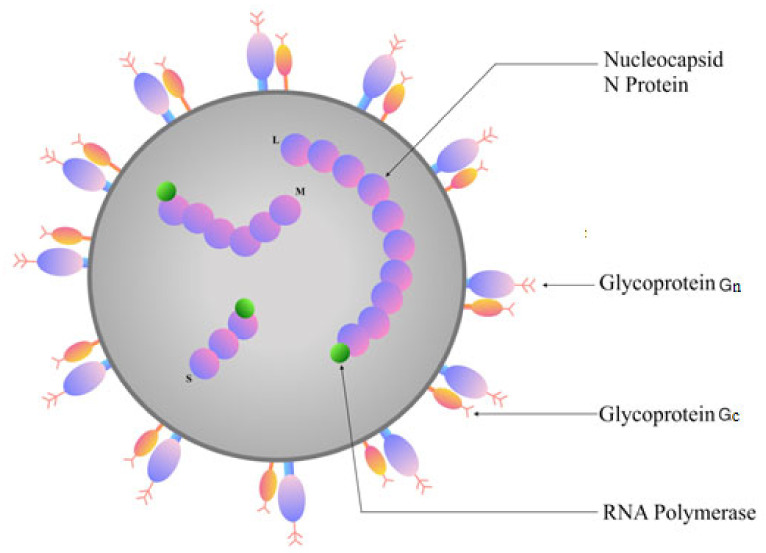
Virion structure of orthohantaviruses. The orthohantavirus virion (ø 80–120 nm) is enveloped. The surface of the virion is surrounded by glycoprotein (Gn and Gc) layer. Inside the virion are three negative-sense segments of single-stranded RNA (the (S) small, (M) medium and (L) large segments).

**Figure 2 pathogens-09-00775-f002:**
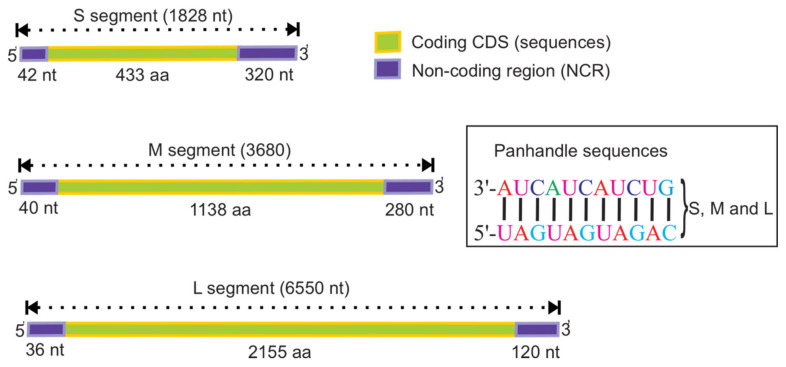
The genome structure of orthohantaviruses, based on Puu/Kazan strain, accession numbers, Z84204, Z84205, and EF405801 for the S, M, and L segments, respectively. The S segment encodes for 433, M-1138, and L-2155 aa. The ends of the segments contain three trinucleotide repeats at the 3′- and 5′-terminal (5′ UAGUAGUAG), which form a panhandle like structure, and it is suggested they are involved in the regulation of viral transcription and replication.

**Figure 3 pathogens-09-00775-f003:**
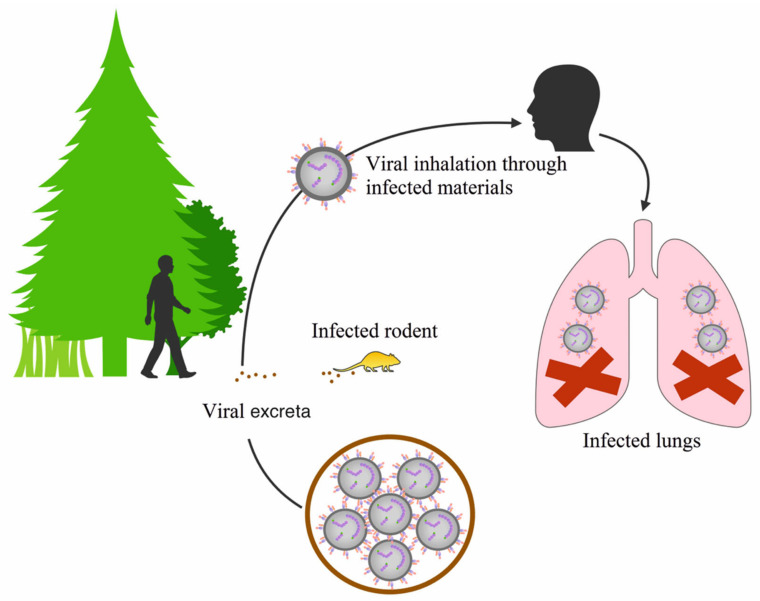
Schematic presentation of orthohantavirus infection cycle. As the rodent population densities increase, the spread of orthohantaviruses among rodents increases proportionally. In rodent and or insectivore populations, orthohantaviruses are horizontally transmitted through aggressive behavior and exposure to aerosolized contaminated droppings. Usually, humans are considered to be the dead-end hosts of orthohantaviruses and become infected by breathing aerosolized rodent excreta containing the virus.

**Figure 4 pathogens-09-00775-f004:**
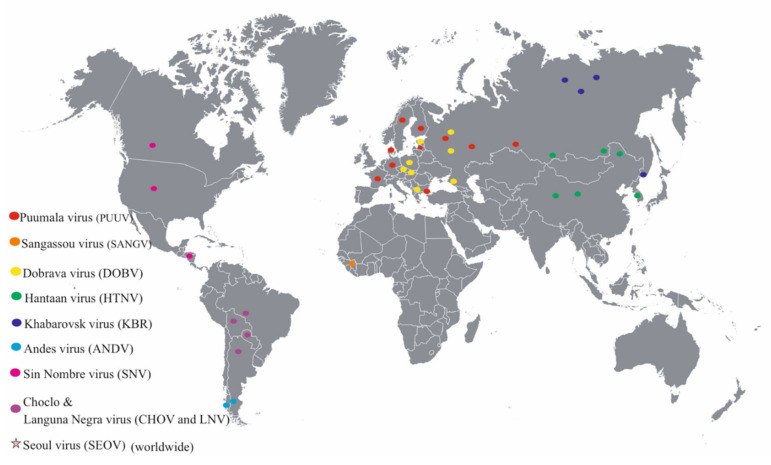
Geographical distribution of human associated pathogenic orthohantaviruses. Seoul orthohantavirus has been detected worldwide.

**Figure 5 pathogens-09-00775-f005:**
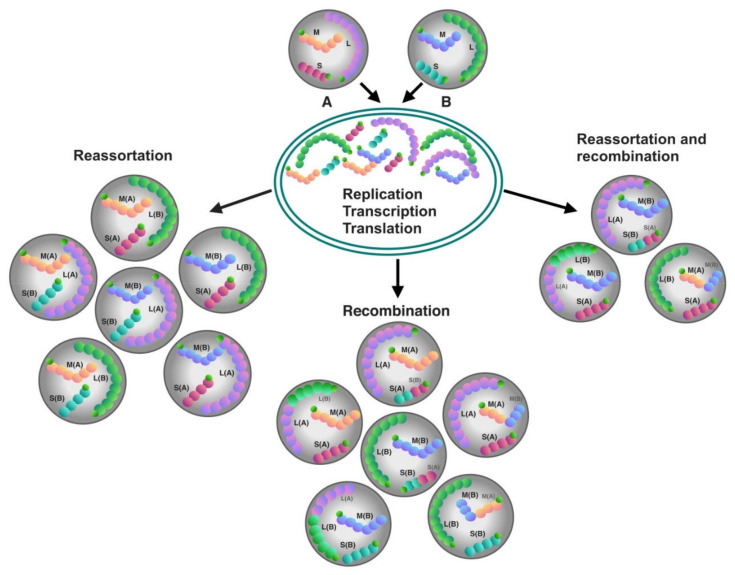
Schematic presentation of recombination and reassortment processes in the tri-segmented orthohantaviruses resulting from the co-infection of a cell by two parental viral strains A and B. The S, M, and L capital letters inside the parental virion strains stands for the S (small), M (medium), and L (large) genomic segments. Parenthesized A and B represent origin of the given segment from one of the two parents. The reassortment and recombination events frequently occur naturally in nature and can also be shown in in vitro experiments.

**Figure 6 pathogens-09-00775-f006:**
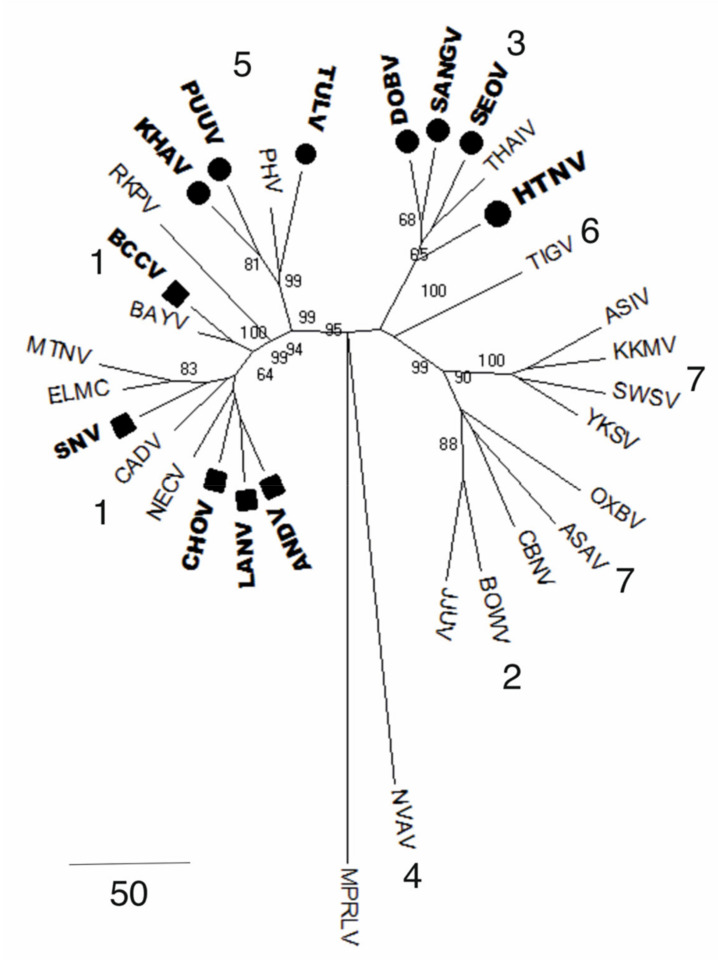
Phylogenetic tree of rodent, shrew, and bat orthohantaviruses based on the full S segment. All PCR confirmed orthohantaviruses associated with human pathogens are marked in bold (HFRS; dark circle and HPS; dark square) and those not associated with the infection. (1) *Sigmodontinae-borne* orthohantaviruses (*Andes*, ANDV; *Laguna Negra*, LANV; *Sin Nombre*, SNV; Montano, MTNV; *Bayou*, BAYV; Black Creek Canal, BCCV; *Choclo*, CHOV; Cano Delgadito, CADV, Maporal, MPRLV, Nicocli, NECV). (2) *Crocidura* orthohantaviruses (Bowe BOWV). (3) *Murinae-borne* orthohantaviruses (*Hantaan*, HTNV; *Dobrava-Belgrade*, DOBV; *Seoul*, SEOV; Thailand, THAIV; Cao Bang, CBNV; Sangassou, SANGV). (4) *Crocidurinae* orthohantaviruses (Nova, NVAV). (5) *Arvicolinae-borne* orthohantaviruses (*Tula*, TULV; *Puumala, PUUV*; Prospect Hill, PHV). (6) *Cricetidae* orthohantaviruses (Rockport, RKPV; Khabarovsk, KHAV; EL Moro Canyon, ELMC; Tigray, TIGV). (7) *Soricinae* orthohantaviruses (Asama, ASAV; Asikkala, ASIV; Jeju, JJUV; Kenkeme, KKMV; Seewis, SWSV; Oxbow, OXBV; Yakeshi, YKSV).

**Table 1 pathogens-09-00775-t001:** Puumala orthohantavirus genetic lineages and geographical location.

	Genetic Lineage (Designation)	Geographical Location	Reference
1	Central European (CE)	France, Belgium, Germany, Slovakia, Netherland	[19,20,21,23,27,28,29]
2	Alpe-Adrian (ALAD)	Austria, Slovenia, Croatia, Hungary	
3	Danish (DAN)	Denmark	
4	South Scandinavian (S-SCAN)	Norway, Southern Sweden	
5	North Scandinavian (N-SCAN)	Northern Sweden	
6	Finnish (FIN)	Finland, Russian Karelia and Western Siberia	
7	Russian (RUS)	Central Russia, Estonia, Latvia	
8	Latvian (LAT)	Latvia, North-East Poland	[18,22]

**Table 2 pathogens-09-00775-t002:** Geographical distribution and reservoir of orthohantaviruses.

Orthohantavirus (Diseases)	Rodents, Natural Reservoir	Area	Reference
*Dobrava-Belgrade* (HFRS)	*Apodemus flavicollis*	Europe (Balkan), European part of Russia	[7]
*Hantaan* (HFRS)	*Apodemus agrarius*	Eastern Asia	
*Khabarovsk* (HFRS)	*Microtus fortis*	Asia (Siberia, Far East of Russia)	
*Puumala* (HFRS)	*Myodes glareolus*	Western, Central and Northern Europe, European part of Russia	[58]
*Sangassou* (HFRS)	*Hylomyscus alleni*	Africa (Guinea)	[68,69]
*Seoul* (HFRS)	*Rattus norvegicus*	Worldwide	[70]
*Tula* (HFRS)	*Microtus arvalis*	Europe	[62]
*Andes* (HPS)	*Oligoryzomys longicaudatus*	South America (Argentina, Chile)	[71]
*Black Creek Canal* (HPS)	*Sigmodon hispidus*	USA	[72]
*Choclo* (HPS)	*Zygodontomys* *brevicauda*	South America (Colombia, French Guiana and Panama)	[73,74]
*Laguna Negra* (HPS)	*Calomys callidus, Akodon simulator, Calomys laucha*	South America (Bolivia, Argentina and Paraguay; parts of Brazil)	[75]
*Sin Nombre* (HPS)	*Peromyscus maniculatus*	Northern America	[71]
*Asama* (unknown)	*Urotrichus* *talpoides*	Asia (Japan)	[76]
*Asikkala* (unknown)	*Sorex minutus*	Europe (Czech Republic, Finland, Germany and Slovakia)	[77]
*Bowe* (unknown)	*Crocidura* *douceti*	Africa (Guinea)	[78]
*Cano Delgadito* (unknown)	*Sigmodon alstoni*	South America (Venezuela)	[79]
*Cao Bang* (unknown)	*Anourosorex* *squamipes*	Asia (China and Vietnam)	[80,81]
*El Moro Canyon* (unknown)	*Reithrodontomys* *megalotis*	USA and Mexico	[82,83]
*Jeju* (unknown)	*Crocidura* *shantungensis*	Asia (Korea)	[76]
*Kenkeme* (unknown)	*Sorex roboratus*	Russia (Siberia)	[84]
*Luxi* (unknown)	*Eothenomys miletus*	Asia (China)	[85]
*Maporal* (unknown)	*Sigmodon alstoni*	South America (Venezuela)	[79]
*Montano* (unknown)	*Peromyscus aztecus*	Northern America (Mexico)	[86]
*Oxbow* (unknown)	*Neurotrichus* *gibbsii*	USA	[87]
*Prospect Hill virus* (unknown)	*Microtus pennsylvanicus*	USA (Maryland)	[88]
*Rockport* (unknown)	*Scalopus* *aquaticus*	USA	[89]
*Seewis* (unknown)	*Sorex* *daphaenodon*	Russia (Siberia)	[90]
*Tatenale* (unknown)	*Microtus agrestis*	United Kingdom (England).	[60,61]
*Tigray* (unknown)	*Stenocephalemys* *albipes*	Africa (Ethiopia)	[91]
*Thailand* (unknown)	*Bandicota indica*	Asia (Thailand)	[92,93]
*Yakeshi* (unknown)	*Sorex isodon*	Asia (China)	[80]
*Necocli, Fusong, Fugong, Dabieshan,* *Bruges*	unknown	unknown	[8]

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
