# Peer review of "Orthohantaviruses, Emerging Zoonotic Pathogens"

_pathogens, 2020, doi:10.3390/pathogens9090775_

Round 1

Reviewer 1 Report

Summary:

In this manuscript, Kabwe et al. provide a thorough review of orthohantaviruses—their geographical distribution, disease association, and genetic diversity. A specific focus is on genomic structure and mechanisms mediating genomic diversity and virus evolution, including recombination, reassortment, and mutation rate. Although orthohantaviruses in general are considered, the authors generally focus on Old World viruses in Eurasia, with a specific emphasis on Puumala virus.

Broad Comments:

  • This manuscript takes an approach that has not previously been done in the current review literature, making it a valuable addition to the field. In general, this review is easy to read and follow. Some parts of the text and figures would benefit from revision for clarity of presentation. Some sections in the latter half (particularly the genetic diversity of Puumala virus) at times feels like a list in paragraph form. These may benefit from additional context or discussion.
  • In the Abstract, the authors state that they anticipate that this review will “contribute to the development of Public Health strategies to improve the prevention and spread of HFRS”. However, the review does not address this directly. Some relevance is implied (i.e., the impact of genetic diversity on molecular testing), but in general the connection to public health is lacking. The review would benefit from an additional section addressing the implications of orthohantavirus diversity for public health, as well as a discussion of the types of strategies the authors envision. In addition to addressing this specific issue, it could also add helpful context for the discussion of genetic diversity in general.

Specific Comments:

  • Line 52: Mentioning SARS-CoV-2 is understandable given the current pandemic. However, the current known mechanisms of spread for SARS-CoV-2 (human to human) compared to orthohantaviruses (humans as dead-end hosts) appears to be substantially different, making this feel like a weak connection.
  • Lines 59-62: I find these sentences a little confusing. It’s not clear to me whether the authors are referring to different groups of bank voles (migrating from different areas to a common area), or whether completely different rodents are involved.
  • Line 68 and Table 1: The references mentioned in the text (17-18) versus the table (17, 21, 23-28) do not appear to match. Why is that the case?
  • Line 75: What is the pathogenesis in rodents? Is it the same across all strains/hosts, or is it variable, just without any correlation?
  • Line 100: “L” should refer to “large” rather than “significant”.
  • Lines 115-116: There appears to be a typo with the parentheses in this sentence. Some re-wording could help as well.
  • Figure 2: Is there a specific virus that the segment sizes are based on? There appear some typos in this figure (“H” and “a.o.”—supposed to be “nt” and “aa”?). Some of the lettering appears to be stretched/compressed, and the font sizes don’t all match for the labels.
  • Lines 126-127: It’s unclear to me why this is a new paragraph.
  • Table 2: There is a gap row in the table. Is this intentional? If so, what is the purpose? On the bottom row, “ICTV” is listed in the animal reservoir and Area columns. Should these instead be “unknown”?
  • Line 180: Missing period after the reference.
  • Lines 183-184: This sentence seems redundant, as this information is already covered in the previous paragraphs.
  • Lines 180-182 and Figure 4: The text says that SEOV “is now spreading”, while Figure 4 says that it is “worldwide”. Could the authors clarify? Perhaps SEOV could be added to the map with arrows showing where active spread is occurring. There also appears to be some mismatched font sizes in the figure.
  • Lines 198-214: The authors mention cauliflower mosaic virus as a classic example of recombination, including numerical estimates of the frequency. However, the virus is in a completely different family with a much different genomic structure. Based on this, it seems like orthohantaviruses are likely to have a much different recombination rate. Have any attempts been made to quantify recombination frequency in Hantaviridae? The current wording of the paragraphs could lead readers to conclude that orthohantaviruses recombine at a similar frequency to cauliflower mosaic virus, which may be misleading. If no quantitative studies have been performed, it may be helpful to explicitly communicate this at the beginning of the Hantaviridae paragraph.
  • Figure 6: This figure should be improved. Since abbreviations are provided in the legend, those could be used instead of full names to clean up the look. The figure and legend also don’t appear to completely match (Nova virus is in the legend but appears to be missing from the figure). There is a branch neighboring Asikkala that appears to have been erased/lost. It would also be helpful to add visual labels to identify the different virus-host associations, such as symbols, numbers, boxes, or color-coding.

Author Response

Rebuttal letter

[Pathogens] Manuscript ID: pathogens-929209: Orthohantaviruses, Emerging Zoonotic Pathogens

We would like to thank the reviewers for their very constructive comments and detailed suggestions for the manuscript. We believe that the comments have identified important areas which required improvement. We have revised the text, incorporating all the suggestions made by the three reviewers. After completion of the suggested edits, the revised manuscript has benefitted from an improvement in the overall presentation and clarity. Below, you will find a point-by-point description of how each comment was addressed in the manuscript. The changes are highlighted yellow in the manuscript.

Responses to reviewers’ comments: Original reviewer comments in italics, with responses in regular typeface.

REVIEWER 1

Broad Comments:

  • This manuscript takes an approach that has not previously been done in the current review literature, making it a valuable addition to the field. In general, this review is easy to read and follow. Some parts of the text and figures would benefit from revision for clarity of presentation. Some sections in the latter half (particularly the genetic diversity of Puumala virus) at times feels like a list in paragraph form. These may benefit from additional context or discussion.

The authors thank the reviewer for their kind words. Sections throughout have been edited and changed, as have some of the figures so we feel it is now an improved version of the manuscript.

  • In the Abstract, the authors state that they anticipate that this review will “contribute to the development of Public Health strategies to improve the prevention and spread of HFRS”. However, the review does not address this directly. Some relevance is implied (i.e., the impact of genetic diversity on molecular testing), but in general the connection to public health is lacking. The review would benefit from an additional section addressing the implications of orthohantavirus diversity for public health, as well as a discussion of the types of strategies the authors envision. In addition to addressing this specific issue, it could also add helpful context for the discussion of genetic diversity in general.

The authors have reworded to the abstract to not overstate the importance of the information presented. It was not the authors’ intent to set out the potential strategies, just highlight some of the known factors that potentially increase the density of rodent populations and therefore increase the risk of viral spillover. The authors have expanded Section 3 to elaborate on some of these themes (Lines 151-160)

Specific Comments:

  • Line 52: Mentioning SARS-CoV-2 is understandable given the current pandemic. However, the current known mechanisms of spread for SARS-CoV-2 (human to human) compared to orthohantaviruses (humans as dead-end hosts) appears to be substantially different, making this feel like a weak connection.

The authors agree and the sentence has been deleted

  • Lines 59-62: I find these sentences a little confusing. It’s not clear to me whether the authors are referring to different groups of bank voles (migrating from different areas to a common area), or whether completely different rodents are involved.

The authors apologize for any confusion. ‘Rodent’ has been changed to bank vole

  • Line 68 and Table 1: The references mentioned in the text (17-18) versus the table (17, 21, 23-28) do not appear to match. Why is that the case?

This was an oversight when preparing the manuscript. The references have now been corrected

  • Line 75: What is the pathogenesis in rodents? Is it the same across all strains/hosts, or is it variable, just without any correlation?

The authors agree that orthohantaviruses discussed do not cause disease in the rodents; therefore, the sentence was changed to reflect and clarify the meaning (Lines 78-80).  

  • Line 100: “L” should refer to “large” rather than “significant”.

The authors have made the typographical change

  • Lines 115-116: There appears to be a typo with the parentheses in this sentence. Some re-wording could help as well.

This section has been reworded in response to Reviewer 2

  • Figure 2: Is there a specific virus that the segment sizes are based on? There appear some typos in this figure (“H” and “a.o.”—supposed to be “nt” and “aa”?). Some of the lettering appears to be stretched/compressed, and the font sizes don’t all match for the labels.

The figure has been updated

  • Lines 126-127: It’s unclear to me why this is a new paragraph.

Answer: Agreed, it was mistakenly joined with chapter 4. This is supposed to be the end of chapter 3. The chapter 4 was mistakenly deleted and combined with chapter 3. It was reversed

  • Table 2: There is a gap row in the table. Is this intentional? If so, what is the purpose? On the bottom row, “ICTV” is listed in the animal reservoir and Area columns. Should these instead be “unknown”?

The table has been tidied up with the row deleted and ICTV was replaced by “unknown”.

  • Line 180: Missing period after the reference.

The authors have made the typographical change

  • Lines 183-184: This sentence seems redundant, as this information is already covered in the previous paragraphs.

The sentence has been removed

  • Lines 180-182 and Figure 4: The text says that SEOV “is now spreading”, while Figure 4 says that it is “worldwide”. Could the authors clarify? Perhaps SEOV could be added to the map with arrows showing where active spread is occurring. There also appears to be some mismatched font sizes in the figure.

Now line 210. The sentence has been reworded to “has now spread”.

  • Lines 198-214: The authors mention cauliflower mosaic virus as a classic example of recombination, including numerical estimates of the frequency. However, the virus is in a completely different family with a much different genomic structure. Based on this, it seems like orthohantaviruses are likely to have a much different recombination rate. Have any attempts been made to quantify recombination frequency in Hantaviridae? The current wording of the paragraphs could lead readers to conclude that orthohantaviruses recombine at a similar frequency to cauliflower mosaic virus, which may be misleading. If no quantitative studies have been performed, it may be helpful to explicitly communicate this at the beginning of the Hantaviridae paragraph.

The text has been updated and a sentence (Line 236-238) has been added: “In contrast to other viruses, the frequency of genome recombination in the hantaviridae family is not fully understood.

  • Figure 6: This figure should be improved. Since abbreviations are provided in the legend, those could be used instead of full names to clean up the look. The figure and legend also don’t appear to completely match (Nova virus is in the legend but appears to be missing from the figure). There is a branch neighboring Asikkala that appears to have been erased/lost. It would also be helpful to add visual labels to identify the different virus-host associations, such as symbols, numbers, boxes, or color-coding.

The authors thank the reviewer for valuable suggestions on how to improve this figure. The authors have added the full names to the legend, checked and corrected any missing values. Also, the figure was improved by using the acronym representing the name of the virus and put in bold to differentiate pathogenic from non-pathogenic viruses.

Reviewer 2 Report

This review by Kabwe et al describes the distribution of orthohantavirus in Eurasia with a focus on Puumala virus and its genetic diversity evolution. The chapters dealing with the geographical distribution, hosts and the different biological mechanisms behind the great genomic diversity of these viruses are fairly well explained.

However, I do have several major concerns on this review:

In my opinion some important elements for such a review are missing.

- First, in a review on Puumala virus diversity and evolution I would suggest to add some elements on the current knowledges on the different domains of the proteins, at least for the N (Location and diversity level of the different domain like the hypervariable and major antigenic domain).

- Some orthohantaviruses possess a NSs protein (like PUUV), this should be mentioned in chapter 2 and 7.

- The knowledges on evolutionary forces (positive and/or negative selection) involved in the evolution of orthohantaviruses are very little described and would need more details.

- I would appreciate more elements on the current geographical extensions of these viruses, what would justify their “emerging” character and would give tools for “the development of Public Health strategies”.

- In my opinion, Reference 37 is not appropriate for the statement “it appears that the nucleotide sequence diversity rate is linked to the geographic location of the bank vole population” (line 350). More recent publications as the work of Binder et al, 2020 are not discussed in this paragraph and should be.

- Moreover chapter 7 is difficult to follow as it is supposed talk about Puumala virus but lines 315-326 are about other orthohantaviruses… It is also difficult to understand the segment considered in some part of the text (lines 318-319 for example). This part needs some clarifications.

- The suggestion that PUUV is the most variable virus within the orthohantavirus genus (lines 374-376) needs references or objective comparisons. But as it will also depend of the number of sequences available for each different virus, it should be nuanced!

Other comments:

- Lines 50-51: the comparison between the potential risk of spreading of SARS-CoV-2 and a virus like SEOV with no human-to-human transmission seems inappropriate to me.

- Lines 55 (and others): PUUV is responsible for an attenuate form of HFRS (nephropathia epidemica), it should be specified.

- Table 1: Netherland should be added in the geographical location of CE lineage.

- Figure 3:  “feces” should be replaced with “excreta”

- Lines 226-228: Razzauti et al have shown that reassortants exist in the wild but are transients with no competitive advantages to the parental strains. This part needs some precisions.

- Lines 230-234: It is not clear if the authors discuss of PUUV or orthohantaviruses in general, please clarify these sentences.

- Figure 6: I don’t understand the sentence “All serologically confirmed orthohantaviruses associated with human pathogens and those not associated with the infection” in the context.

Author Response

Rebuttal letter

[Pathogens] Manuscript ID: pathogens-929209: Orthohantaviruses, Emerging Zoonotic Pathogens

We would like to thank the reviewers for their very constructive comments and detailed suggestions for the manuscript. We believe that the comments have identified important areas which required improvement. We have revised the text, incorporating all the suggestions made by the three reviewers. After completion of the suggested edits, the revised manuscript has benefitted from an improvement in the overall presentation and clarity. Below, you will find a point-by-point description of how each comment was addressed in the manuscript. The changes are highlighted yellow in the manuscript.

Responses to reviewers’ comments: Original reviewer comments in italics, with responses in regular typeface.

REVIEWER 2

In my opinion some important elements for such a review are missing.

- First, in a review on Puumala virus diversity and evolution I would suggest to add some elements on the current knowledges on the different domains of the proteins, at least for the N (Location and diversity level of the different domain like the hypervariable and major antigenic domain).

The authors have added the requested detail at lines 110-113 and lines 380-384

- Some orthohantaviruses possess a NSs protein (like PUUV), this should be mentioned in chapter 2 and 7.

The authors have added sentences at lines 106-108 and lines 400-403.

- The knowledges on evolutionary forces (positive and/or negative selection) involved in the evolution of orthohantaviruses are very little described and would need more details.

The authors have added sentences at lines 325-328

- I would appreciate more elements on the current geographical extensions of these viruses, what would justify their “emerging” character and would give tools for “the development of Public Health strategies”.

The authors thank the reviewer for identifying this area to expand on. Section 3 has been expanded to elaborate on these themes (Lines 151-160)

- In my opinion, Reference 37 is not appropriate for the statement “it appears that the nucleotide sequence diversity rate is linked to the geographic location of the bank vole population” (line 350). More recent publications as the work of Binder et al, 2020 are not discussed in this paragraph and should be.

This sentence has been reworded and extra references added to improve the argument (Lines 393-396). The authors also agree that the addition of Binder et al enhances the manuscript and has been added (Lines 400-403)

- Moreover chapter 7 is difficult to follow as it is supposed talk about Puumala virus but lines 315-326 are about other orthohantaviruses… It is also difficult to understand the segment considered in some part of the text (lines 318-319 for example). This part needs some clarifications.

We thank the reviewer for point out this slightly confusing section. This is now section 6 and subdivided into other orthohantaviruses and PUUV. The wording has also been revised to aid understanding.

- The suggestion that PUUV is the most variable virus within the orthohantavirus genus (lines 374-376) needs references or objective comparisons. But as it will also depend of the number of sequences available for each different virus, it should be nuanced!

This is now Line 428-430 and references have been added. The sentence has also been changed to reflect that it could be sampling bias

Other comments:

- Lines 50-51: the comparison between the potential risk of spreading of SARS-CoV-2 and a virus like SEOV with no human-to-human transmission seems inappropriate to me.

The authors agree and the sentence has been deleted

- Lines 55 (and others): PUUV is responsible for an attenuate form of HFRS (nephropathia epidemica), it should be specified.

This has been added to the text.

- Table 1: Netherland should be added in the geographical location of CE lineage.

We thank the reviewer for this spotting this oversight and Netherlands has now been added

- Figure 3:  “feces” should be replaced with “excreta”

The authors have made this typographical change

- Lines 226-228: Razzauti et al have shown that reassortants exist in the wild but are transients with no competitive advantages to the parental strains. This part needs some precisions.

The sentence has been changed to reflect the unknown nature as to whether the reassortants can out-compete the parental strains

- Lines 230-234: It is not clear if the authors discuss of PUUV or orthohantaviruses in general, please clarify these sentences.

The authors have added ‘PUUV’ and ‘other’ at Lines 266-267 to avoid confusion

- Figure 6: I don’t understand the sentence “All serologically confirmed orthohantaviruses associated with human pathogens and those not associated with the infection” in the context.

This has been changed and the figure updated in line with Reviewer 1’s comments

Reviewer 3 Report

Major comments:

  1. I noticed there are listed authors from USA and UK. The manuscript should be reviewed by native English again.
  2. The title of this paper is about Orthohanta viruses, however, authors did talk a lot about PUUV. Is it not consistent with the title. 
  3. At the end of paper, one large paragraph is for PUUV. It is better to put it in the introduction part.

Minor comments:

Line 27: Public Health strategies to improve the prevention and spread of HERS. What does it mean?

Lne 20: PUUV stands for Puumala orthohantaviruses not Puumala.

Line 106: The paper needs to address more details about different proteins function based on references (3, 37, 39, 40 and 41).

Author Response

Rebuttal letter

[Pathogens] Manuscript ID: pathogens-929209: Orthohantaviruses, Emerging Zoonotic Pathogens

We would like to thank the reviewers for their very constructive comments and detailed suggestions for the manuscript. We believe that the comments have identified important areas which required improvement. We have revised the text, incorporating all the suggestions made by the three reviewers. After completion of the suggested edits, the revised manuscript has benefitted from an improvement in the overall presentation and clarity. Below, you will find a point-by-point description of how each comment was addressed in the manuscript. The changes are highlighted yellow in the manuscript.

Responses to reviewers’ comments: Original reviewer comments in italics, with responses in regular typeface.

Major comments:

- I noticed there are listed authors from USA and UK. The manuscript should be reviewed by native English again.

The manuscript and rebuttal letter have been reviewed and edited by native English speakers.

The title of this paper is about Orthohantaviruses, however, authors did talk a lot about PUUV. Is it not consistent with the title. 

The authors acknowledge the reviewers point that part of the manuscript is very ‘PUUV-centric’. This is by design as it is the most studied orthohantavirus, but the authors have tried to put it into context with other orthohantavirus family members. It is our feeling that a title concentrating on PUUV would be underselling the other aspects of the paper.

At the end of paper, one large paragraph is for PUUV. It is better to put it in the introduction part.

The authors acknowledge it is a large paragraph on PUUV, but as it is specifically on the genetic diversity of PUUV, we feel that it warrants being a section on its own. The section has also been subdivided as it compares the genetic diversity of other orthohantaviruses.

Minor comments:

Line 27: Public Health strategies to improve the prevention and spread of HERS. What does it mean?

This line has been rewritten for clarity

Lne 20: PUUV stands for Puumala orthohantaviruses not Puumala.

We thank the reviewer for highlighting this typographical error and it has been corrected.

Line 106: The paper needs to address more details about different proteins function based on references (3, 37, 39, 40 and 41).

The functions of the proteins have been expanded on in Lines 106-117

Round 2

Reviewer 2 Report

The authors significantly improved their manuscript and responded to the points I had raised.

From my point of view the manuscript is now acceptable for publication.

I have just notified 3 minor points that the authors wrote they have modified but that I don't see in the new version.

1) Figure 3 has not been modified ("viral faeces" should be changed to "excreta")

2) The sentence comparing SEOV to SARS-Cov-2 (lines 52-53) has not been removed and should be.

3) line 396, "based on Orlean PUUV strains" should be removed as it's general for PUUV. 

Author Response

We would like to thank the reviewer again for his constructive comments and detailed suggestions for the manuscript. We believe that the comments have identified important areas which required improvement. We have revised the text, incorporating all the suggestions made by the reviewer. After completion of the suggested edits, the revised manuscript has benefitted from an improvement in the overall presentation and clarity. Below, you will find a point-by-point description of how each comment was addressed in the manuscript. The changes are highlighted yellow in the manuscript.

1) Figure 3 has not been modified ("viral faeces" should be changed to "excreta")

The authors have made this typographical change to the figure

2) The sentence comparing SEOV to SARS-Cov-2 (lines 52-53) has not been removed and should be.

The authors agree and the sentence has been deleted

3) line 396, "based on Orlean PUUV strains" should be removed as it's general for PUUV.

The authors agree and the sentence has been modified accordingly (based on Orlean PUUV strains was deleted)